# Application of a Multiplex Polymerase Chain Reaction Test for Diagnosing Bacterial Enteritis in Children in a Real-Life Clinical Setting

**DOI:** 10.3390/children8070538

**Published:** 2021-06-24

**Authors:** Hyun-Woo Lee, Seung-Beom Han, Jung-Woo Rhim

**Affiliations:** 1Department of Pediatrics, College of Medicine, The Catholic University of Korea, Seoul 06591, Korea; zzangwoo91@naver.com (H.-W.L.); jwrhim@catholic.ac.kr (J.-W.R.); 2The Vaccine Bio Research Institute, College of Medicine, The Catholic University of Korea, Seoul 06591, Korea; 3Department of Pediatrics, Daejeon St. Mary’s Hospital, The Catholic University of Korea, Daejeon 34943, Korea

**Keywords:** gastroenteritis, polymerase chain reaction, bacteria, child

## Abstract

This study aimed to determine the subjects for bacterial multiplex polymerase chain reaction (mPCR) testing and to interpret the mPCR test results based on patients’ clinical symptoms and diagnoses. The medical records of 710 pediatric patients who underwent a bacterial mPCR test were retrospectively reviewed. Clinical characteristics and mPCR test results were compared between patients with positive (*n* = 199) and negative mPCR test results (*n* = 511) and between patients with invasive pathogens (*n* = 95) and toxigenic pathogens (*n* = 70). Positive mPCR test results were significantly associated with older age (*p* < 0.001), diagnosis of acute gastroenteritis (*p* = 0.021), presence of hematochezia (*p* < 0.001), and absence of cough (*p* = 0.004). The diagnosis of acute gastroenteritis (*p* = 0.003), presence of fever (*p* = 0.027) and diarrhea (*p* = 0.043), and higher C-reactive protein levels (*p* = 0.025) were significantly associated with the identification of invasive pathogens in patients with positive mPCR test results. Thus, selective bacterial mPCR testing should be performed based on the patients’ clinical symptoms and diagnoses, and the results should be interpreted in consideration with identified pathogens.

## 1. Introduction

Infectious acute gastroenteritis (AGE) is caused by various viruses, bacteria, and parasites [1]. Each of these pathogens causes various gastrointestinal (GI) symptoms and some of them induce similar GI symptoms [1]. Therefore, determining the causative pathogen on the basis of only patient’s symptoms is difficult. Moreover, performing various laboratory studies targeting each GI pathogen in each patient is time consuming, laborious, and thus, not cost-effective [2]. To improve the use of multiple test runs, a syndromic diagnosis has been used to identify the pathogens of infectious AGE, and multiplex polymerase chain reaction (mPCR) testing is representative [2]. Although there have been several studies on mPCR tests to identify GI pathogenic bacteria using stool samples [3,4,5,6,7,8,9,10,11], most of them evaluated the performance of mPCR tests on stool samples as requested by clinicians, without considering the patients’ GI symptoms or clinical diagnoses [7,8,9,10,11]. However, vomiting and diarrhea, the major symptoms of infectious AGE, can be caused by infections in systems other than the GI tract, such as the respiratory tract, urinary tract, and central nervous system, as well as by toxins, medications, and non-infectious neurological, endocrine, and allergic diseases [12,13]. Prolonged excretion of GI pathogens colonizing the GI tract after an improvement in symptomatic or asymptomatic infection might be detected by performing an mPCR test [2,4]. In a real-life clinical setting, it is difficult to determine whether unexpected GI pathogens detected after conducting an mPCR test are the causative agents of infectious AGE or bystanders found in patients with GI symptoms accompanied by diseases other than infectious AGE. This situation may confound the selection of appropriate treatments and infection control measures. Therefore, a bacterial mPCR test should be performed selectively in patients with GI symptoms consistent with bacterial AGE, and its results should be interpreted based on patients’ symptoms and clinical diagnoses.

This retrospective study aimed to determine the subjects for bacterial mPCR testing and to interpret the mPCR test results based on patients’ clinical symptoms and diagnoses in pediatric patients. These results can be used as a basis for deciding the appropriate application of a bacterial mPCR test and interpretation of its results in pediatric patients with GI symptoms.

## 2. Materials and Methods

### 2.1. Patients and Study Design

Pediatric patients (aged <19 years) who were admitted to the Department of Pediatrics of Daejeon St. Mary’s Hospital (Daejeon, Korea) between June 2015 and August 2019 and underwent bacterial mPCR testing were considered for this study. Patients with GI symptoms that developed >48 h after admission or ≤48 h after a previous discharge from the hospital were considered to have hospital-acquired infections and were excluded. Additionally, patients with chronic underlying diseases or chronic diarrhea persisting for ≥2 weeks were excluded. The medical records of the included patients were reviewed retrospectively to collect demographic data including age and sex, clinical data including the patient’s symptoms, clinical diagnoses leading to admission, and fever duration, and laboratory data including complete blood cell count, erythrocyte sedimentation rate (ESR), C-reactive protein (CRP) and electrolyte levels, and liver and kidney function test results. The included patients were divided into two groups based on their mPCR test results: positive and negative mPCR test groups. For the positive mPCR test group, patients were further categorized into invasive pathogen and toxigenic pathogen groups based on the identified bacteria; patients with multiple pathogens were excluded. Pathogenic bacteria identified in a small number of patients (<1% of the whole study population) were not categorized into the invasive pathogen or toxigenic pathogen groups and were excluded from statistical analyses. The collected data and the distribution of bacteria detected by an mPCR test were compared between these patient groups. This study was approved by the Institutional Review Board of Daejeon St. Mary’s Hospital with a waiver for acquiring informed consent (approval no.: DC19RESI0084).

### 2.2. Microbiological Test

Fresh stool samples collected from admitted patients were transported to the Department of Laboratory Medicine as soon as possible. A Ribospin vRD kit (GeneAll Biotechnology, Seoul, Korea) was used in accordance with the manufacturer’s recommendations for DNA extraction, and a bacterial mPCR test was performed using a commercial mPCR test kit (Seeplex^®^ Diarrhea-B1/B2 ACE detection kit, Seegene, Seoul, Korea) in accordance with the manufacturer’s recommendations. This mPCR kit can simultaneously detect 10 types of bacterial pathogens, including *Campylobacter* spp. (*C. jejuni* and *C. coli*), *Salmonella* spp. (*S. enterica* and *S. bongori*), *Shigella* spp. (*S. boydii*, *S. dysenteriae*, *S. flexneri*, and *S. sonnei*), *Clostridium difficile* toxin B, *Clostridium perfringens*, *Vibrio* spp. (*V. cholerae*, *V. parahaemolyticus*, and *V. vulnificus*), *Yersinia enterocolitica*, *Aeromonas* spp. (*A. bivalvium*, *A. hydrophila*, *A. salmonicida*, and *A. sobria*), *Escherichia coli* O157:H7, and verotoxin-producing *E. coli* (VTEC) [11].

### 2.3. Statistical Analysis

To compare the positive and negative mPCR test groups and the invasive pathogen and toxigenic pathogen groups, chi-square and Mann–Whitney U tests were used for categorical and continuous data, respectively. Multivariate analyses of significant factors identified in the univariate analyses were performed using a binary logistic regression test to determine those associated with a positive mPCR test result and the identification of invasive GI pathogens by a bacterial mPCR test. The SPSS 21 program (IBM Corporation, Armonk, NY, USA) was used for statistical analyses. The threshold of statistical significance was defined as a *p*-value of 0.05.

## 3. Results

During the study period, bacterial mPCR tests were performed in 814 pediatric patients. Among them, 49, 43, and 12 patients were excluded due to probable hospital-acquired infections, chronic underlying diseases, and chronic diarrhea, respectively. The remaining 710 patients included in this study had a median age of 5 years (interquartile range [IQR]: 1–9), and 400 (56.3%) were men. For 199 (28.0%) patients, at least one of the pathogenic bacteria was identified by performing a bacterial mPCR test (Table 1). 

Two and three types of bacteria were co-detected in 19 (9.5%) and two (1.0%) patients, respectively. Co-detection of bacteria was reported in 12.5% of *Campylobacter* infections, 15.4% of *C. difficile* infections, 15.2% of *Salmonella* infections, 36.6% of *C. perfringens* infections, and 47.1% of *Aeromonas* infections. As a single pathogen, *Campylobacter* spp., *C. difficile*, *Salmonella* spp., and *C. perfringens* were the most frequently identified pathogens (Table 1). Excluding rarely identified pathogens, the most frequently identified four pathogens were categorized into invasive and toxigenic pathogens based on their pathogenic mechanisms [14]. Patients with *Campylobacter* spp. and *Salmonella* spp. were categorized in the invasive pathogen group (*n* = 95), and those with *C. difficile* and *C. perfringens* were categorized in the toxigenic pathogen group (*n* = 70). The remaining bacteria, comprising 6.5% of the positive mPCR test results, were excluded from further statistical analyses.

### 3.1. Comparison between the Positive and Negative mPCR Test Groups

Among the 710 patients, 511 (72.0%) and 199 (28.0%) patients were assigned to the negative and positive mPCR test groups, respectively (Table 2). Older age, diagnosis of AGE, presence of fever and GI symptoms, absence of respiratory symptoms, and high ESR and CRP levels were significantly associated with a positive mPCR test result in the univariate analysis (Table 2).

In the multivariate analysis, a positive mPCR test result was significantly associated with older age (*p* < 0.001), diagnosis of AGE (*p* = 0.021), presence of hematochezia (*p* < 0.001), and absence of cough (*p* = 0.004, Table 3). Among 36 patients diagnosed with non-AGE in the positive mPCR test group, *C. difficile*, *C. perfringens*, and *C. difficile* and *C. perfringens* were identified in 19, seven, and three patients, respectively. *Aeromonas* spp. and *Salmonella* spp. were identified in two patients, and *Campylobacter* spp., VTEC, and *Salmonella* spp. and *C. perfringens* were identified in one patient each.

### 3.2. Comparison between the Invasive Pathogen and Toxigenic Pathogen Groups

The invasive pathogen group consisted of 56 and 39 patients in whom *Campylobacter* spp. and *Salmonella* spp. were identified, respectively. The toxigenic pathogen group consisted of 44 and 26 patients in whom *C. difficile* and *C. perfringens* were identified, respectively. In the univariate analysis, the detection of invasive pathogens was significantly associated with older age, diagnosis of AGE, presence of fever, diarrhea, and abdominal pain, absence of vomiting and respiratory symptoms, and high ESR and CRP levels (Table 4).

In the multivariate analysis, diagnosis of AGE (*p* = 0.003), presence of fever (*p* = 0.027) and diarrhea (*p* = 0.043), and high CRP levels (*p* = 0.025) were significantly associated with the detection of invasive pathogens (Table 5). For the 44 patients with positive results for *C. difficile*, the median age was 3 years (IQR: 1–8), and 14 (31.8%) were aged <2 years. Histories of hospitalization and antibiotic therapy within the previous 2 months were identified in eight (18.2%) and 19 (43.2%) patients, respectively. Metronidazole treatment was administered to 13 (29.5%) patients. AGE was more frequently diagnosed in patients who received metronidazole treatment (11/13, 84.6%) than in those who did not receive metronidazole treatment (14/31, 45.2%; *p* = 0.016). 

## 4. Discussion

In this study, the results of bacterial mPCR tests performed in pediatric patients in a real-life clinical setting were analyzed. Clinical diagnosis of AGE and the presence of fever were significantly associated with a positive result in a bacterial mPCR test and identification of invasive GI pathogens.

Various mPCR tests for GI pathogenic bacteria have been reported to have favorable diagnostic performance [4,5,6,7,8,9,10,11]; however, a few studies have evaluated the performance of these tests based on the patients’ clinical diagnoses [6,15,16]. Viruses rather than bacteria are major causes of infectious AGE, and a specific antibiotic therapy is not urgently required in most cases of community-acquired bacterial AGE in immunocompetent patients [1]. Therefore, a bacterial mPCR test should be performed selectively in patients with suspected bacterial AGE, which would, in turn, increase the diagnostic accuracy of the bacterial mPCR test, reduce associated costs and labor, and avoid unnecessary patient discomfort. Older age was significantly associated with a positive bacterial mPCR test result in the multivariate analysis in this study. However, the exact age with acceptable sensitivity and specificity for the identification of pathogenic bacteria could not be defined. Although the presence of hematochezia and the absence of cough were significantly associated with a positive mPCR test result, only 17.6% and 28.0% of patients in the positive and negative mPCR test groups, respectively, exhibited these symptoms. Therefore, their usefulness in determining subjects for bacterial mPCR testing should be small.

In this study, one-third of the included patients underwent bacterial mPCR testing but were not clinically diagnosed with AGE. Since the diagnosis of AGE was significantly associated with a positive mPCR test result and identification of invasive pathogens that are potential candidates for antibiotic therapy, selective bacterial mPCR testing in patients who are clinically diagnosed with infectious AGE should be encouraged. Recent advancements in microbiological laboratory methods tend to enhance physicians’ dependence on laboratory test results more than patient history and physical examination. However, microbiological tests should be performed according to the presumptive diagnosis made based on the patient’s history and physician’s examination [17]. The results of this study emphasized the importance of thorough history taking and physical examination for making an accurate clinical diagnosis. Some patients may complain of vomiting only when coughing, abdominal pain developed after severe coughing, or loose stool not consistent with diarrhea (<3 episodes in a day). If the primary clinical diagnosis is infections other than AGE, such as respiratory tract and urinary tract infections, bacterial mPCR testing can be omitted. Because respiratory tract infection was the most frequent diagnosis in non-AGE cases, detailed history taking and physical examination to determine whether AGE and respiratory tract infection co-exist or whether respiratory symptoms promote GI symptoms are required. To reduce unnecessary stool examination in patients presenting with concurrent GI and respiratory symptoms, the order of development, trend of severity, and simultaneity of the GI and respiratory symptoms should be considered during clinical diagnosis. Among 243 patients diagnosed with non-AGE, 36 (14.8%) patients showed positive mPCR test results. However, *C. difficile* and *C. perfringens* were identified in 80.6% of them, and *Campylobacter* spp. and *Salmonella* spp., in which antibiotic therapy were potentially required, were identified only in four (0.6% of the whole study population) patients. Selective bacterial mPCR testing in patients clinically diagnosed with AGE could exempted 34.2% of the included patients from expensive testing and increased the positive rate of the mPCR test from 28.0% to 34.9%.

Among patients in whom *C. difficile* was identified, 29.5% received metronidazole treatment. Although some patients with *C. difficile* infection recover without specific antibiotic treatment [18], AGE was diagnosed significantly less in patients who did not receive metronidazole treatment than in those who did. The colonization rate of *C. difficile* is higher in infants and young children than in adolescents and adults, and by 3 years of age, it decreases and becomes similar to that of adults [18]. *C. difficile* testing should not be routinely recommended for infants with diarrhea and for children aged 1–2 years without the exclusion of other causes of diarrhea [19]. Even for children aged ≥2 years, *C. difficile* testing is recommended for children with risk factors or exposure history to *C. difficile* infection [19]. In the positive mPCR test group of this study, the positive rate of *C. difficile* was significantly higher in children aged <2 years than in those aged ≥2 years (51.5% vs. 21.1%, *p* < 0.001). Moreover, approximately one-third of patients in whom *C. difficile* was detected were aged <2 years, and less than half of them had risk factors for *C. difficile* infection. Therefore, the diagnosis of *C. difficile* infection should not be solely dependent on mPCR testing, but should be confirmed using a multistep algorithm that considers patients’ age, symptoms, and risk factors [18]. In a previous study using the same multiplex PCR test kit used in this study, *C. perfringens* was most frequently detected in children with diarrhea; however, confirmatory tests revealed that the PCR tests showed false positive results [11]. A recent meta-analysis reported low sensitivity (31%) and positive predictive value (49%) of nucleic acid amplification tests against culture for diagnosing *C. perfringens*-associated diseases [20]. Considering that *Clostridium* spp. are more likely to exist as intestinal flora than *Campylobacter* spp. and *Salmonella* spp. [19,21,22,23], determining whether the bacteria detected by an mPCR test are true pathogens is more difficult in cases of *Clostridium* spp. than in cases of invasive pathogens. Furthermore, other bacteria were co-detected in 36.6% of the *C. perfringens*-positive patients in this study. Therefore, universal inclusion of *Clostridium* spp. in bacterial mPCR testing seems to be unnecessary for immunocompetent pediatric patients with GI symptoms. Besides, a high number of *C. perfringens* organisms are needed to cause GI symptoms; therefore, the usefulness of quantitative PCR tests for *C. perfringens* should be investigated [20].

As the presence of fever and diarrhea and higher CRP levels were significantly associated with the identification of invasive pathogens, patients complaining of fever with definite diarrhea rather than prominent vomiting and elevated CRP levels could be subjects for bacterial mPCR testing among those clinically diagnosed with AGE. In developed countries including Korea, *Salmonella* spp. and *Campylobacter* spp. are the most common pathogens causing bacterial AGE [15,16,24,25,26], and they were also most frequently detected in this study. Since *Shigella* spp. and *Vibrio* spp. are rarely identified in patients with AGE in developed countries and that no specific treatment is recommended for AGE caused by *Aeromonas* spp. and *Yersinia* spp. [24,25], a selective test in patients with community-acquired AGE would be more appropriate than a universal test for detecting these bacteria. Enteropathogenic and enteroaggregative *E. coli* were considered major pathogens of bacterial AGE in children, although they were not tested in this study [27]. Therefore, an mPCR test targeting only the most common GI pathogens, including *Campylobacter* spp., *Salmonella* spp., diarrheagenic *E. coli*, rotavirus, and norovirus, should be suitable for patients with community-acquired AGE. Additional tests for rare pathogens can be considered in patients with negative primary test results, risk factors for severe AGE, or immunodeficiency.

This study had some limitations. First, this was a retrospective study that included only hospitalized patients. Thus, some patients with mild symptoms of bacterial AGE might have been managed in the outpatient clinic or have not been subjected for mPCR testing during hospitalization, and hence, excluded from this study. However, the observation that one-third of patients who underwent the mPCR test were not diagnosed with AGE and two-thirds of patients with AGE were negative for bacterial pathogens suggests that bacterial mPCR tests were frequently performed in patients presenting GI symptoms in our hospital with a low threshold for performing the test. Second, confirmatory tests for the pathogens most commonly detected by the mPCR test, such as *Campylobacter* spp. and *Clostridium* spp., were not performed. However, as mentioned above, the results of this study suggest the limited reliability of mPCR testing in diagnosing *Clostridium* infection and need for alternative diagnostic tests. Instead, further studies to compare the results of mPCR tests with those of culture tests for *Campylobacter* spp. and *Salmonella* spp. should be performed based on patients’ clinical diagnoses. Finally, the GI pathogenic viruses were not considered in this study. However, patients with viral AGE might have a marginal impact on the results of this study, unless a significant proportion of them were included in the invasive pathogen group and received unnecessary antibiotic treatment. An mPCR test for GI pathogenic viruses (rotavirus, norovirus, adenovirus, and astrovirus) was simultaneously performed in 432 (60.8%) patients using a commercial kit (Seeplex^®^ Diarrhea-V ACE Detection kit, Seegene): Viral pathogens were identified in 112 (25.9%) patients, and only two (1.8%) of them were identified in the invasive pathogen group.

## 5. Conclusions

In conclusion, a bacterial mPCR test should be performed selectively, and its results should be cautiously interpreted considering patients’ clinical symptoms and diagnoses and identified pathogens. Since some of the GI pathogens are rarely detected even in patients with AGE and *C. difficile* seems to be a bystander rather than a causative pathogen in most cases, an mPCR test for a limited number of common GI pathogens should be performed in real-life clinical settings.

## Figures and Tables

**Table 1 children-08-00538-t001:** Distribution of pathogenic bacteria identified by bacterial multiplex polymerase chain reaction testing.

Bacteria	Number of Patients (*n* = 199)
*Campylobacter* spp.	56 (28.1)
*Clostridium difficile*	44 (22.1)
*Salmonella* spp.	39 (19.6)
*Clostridium perfringens*	26 (13.1)
*Aeromonas* spp.	7 (3.5)
Verotoxin-producing *Escherichia coli*	3 (1.5)
*Shigella* spp.	1 (0.5)
*E. coli* O157/H7	1 (0.5)
*Yersinia* spp.	1 (0.5)
*Vibrio* spp.	0 (0.0)
Co-detection	21 (10.6)
*Salmonella* spp. and *C. perfringens*	5 (2.5)
*Salmonella* spp. and *Aeromonas* spp.	1 (0.5)
*Salmonella* spp., *C. perfringens* and *Aeromonas* spp.	1 (0.5)
*Campylobacter* spp. and *C. perfringens*	4 (2.0)
*Campylobacter* spp. and *Aeromonas* spp.	2 (1.0)
*Campylobacter* spp. and *C. difficile*	1 (0.5)
*Campylobacter* spp., *C. difficile* and *Aeromonas* spp.	1 (0.5)
*C. difficile* and *C. perfringens*	5 (2.5)
*C. difficile* and *Shigella spp.*	1 (0.5)

**Table 2 children-08-00538-t002:** Comparison between patients with positive and negative mPCR test results.

Factor	Negative mPCR Test Group (*n* = 511)	Positive mPCR Test Group (*n* = 199)	*p*-Value
Demographic factor			
Male sex	284 (55.6)	116 (58.3)	0.513
Age, years, median (IQR)	4 (1–8)	6 (3–11)	<0.001
Clinical diagnosis			<0.001
AGE	304 (59.5)	163 (81.9)	
Non-AGE	207 (40.5)	36 (18.1)	
URI	79 (15.5)	13 (6.5)	
LRI	47 (9.2)	6 (3.0)	
Other GI disorders	28 (5.5)	5 (2.5)	
FWLS	17 (3.3)	4 (2.0)	
Urinary tract infection	12 (2.3)	1 (0.5)	
Exanthem subitum	9 (1.8)	3 (1.5)	
CNS disorders	7 (1.4)	3 (1.5)	
Others	8 (1.6)	1 (0.5)	
Clinical symptoms			
Fever	383 (75.0)	173 (86.9)	0.001
Vomiting	310 (60.7)	106 (53.3)	0.072
Diarrhea	314 (61.4)	156 (78.8)	<0.001
Abdominal pain	279 (54.7)	144 (72.4)	<0.001
Hematochezia	27 (5.3)	35 (17.6)	<0.001
Cough	143 (28.0)	18 (9.0)	<0.001
Rhinorrhea	147 (28.8)	27 (13.6)	<0.001
Sputum	109 (21.3)	16 (8.0)	<0.001
Sore throat	18 (3.5)	6 (3.0)	0.737
Laboratory finding, median (IQR)			
WBC count, cells/μL	10,450 (7500–13,600)	9900 (7000–13,100)	0.153
ANC, cells/μL	6474 (4046–9840)	7050 (4320–10,350)	0.365
ALC, cells/μL	2085 (1290–3289)	1463 (1068–2496)	<0.001
ESR, mm/h	7 (2–17)	13 (5–21)	<0.001
CRP, mg/dL	1.12 (0.17–3.99)	3.63 (0.78–7.83)	<0.001

mPCR: multiplex polymerase chain reaction; IQR: interquartile range; AGE: acute gastroenteritis; URI: upper respiratory infection; LRI: lower respiratory infection; GI: gastrointestinal; FWLS: fever without localizing signs; CNS: central nervous system; WBC: white blood cell; ANC: absolute neutrophil count; ALC: absolute lymphocyte count; ESR: erythrocyte sedimentation rate; CRP: C-reactive protein.

**Table 3 children-08-00538-t003:** Multivariate analysis for significant factors associated with positive mPCR test results.

Factor	Odds Ratio	95% Confidence Interval	*p*-Value
Age, years	1.099	1.044–1.158	<0.001
Diagnosis of AGE (vs. non-AGE)	1.935	1.105–3.389	0.021
Fever	2.877	1.628–5.082	<0.001
Diarrhea	1.552	0.962–2.504	0.072
Abdominal pain	0.889	0.548–1.442	0.633
Hematochezia	5.011	2.675–9.385	<0.001
Cough	0.203	0.069–0.602	0.004
Rhinorrhea	1.779	0.785–4.027	0.167
Sputum	1.153	0.407–3.270	0.789
ALC, cells/μL	1.000	1.000–1.000	0.619
ESR, mm/h	1.001	0.991–1.010	0.906
CRP, mg/dL	1.033	0.987–1.082	0.156

mPCR: multiplex polymerase chain reaction; AGE: acute gastroenteritis; ALC: absolute lymphocyte count; ESR: erythrocyte sedimentation rate; CRP: C-reactive protein.

**Table 4 children-08-00538-t004:** Comparison between the invasive pathogen and toxigenic pathogen groups.

Factor	Invasive Pathogen Group (*n* = 95)	Toxigenic Pathogen Group (*n* = 70)	*p*-Value
Demographic factor			
Male sex	56 (58.9)	41 (58.6)	0.961
Age, years, median (IQR)	8 (4–12)	5 (1–11)	0.009
Age group			0.009
<2 years	8 (8.4)	18 (25.7)	
2–8 years	43 (45.3)	28 (40.0)	
>8 years	44 (46.3)	24 (34.3)	
Clinical diagnosis			<0.001
AGE	92 (96.8)	44 (62.9)	
Non-AGE	3 (3.2)	26 (37.1)	
URI	1 (1.1)	10 (14.3)	
LRI	0 (0.0)	5 (7.1)	
Other GI disorders	1 (1.1)	3 (4.3)	
FWLS	1 (1.1)	1 (1.4)	
Urinary tract infection	0 (0.0)	1 (1.4)	
Exanthem subitum	0 (0.0)	2 (2.9)	
CNS disorders	0 (0.0)	3 (4.3)	
Others	0 (0.0)	1 (1.4)	
Clinical symptoms			
Fever	94 (98.9)	51 (72.9)	<0.001
Vomiting	44 (46.3)	45 (64.3)	0.022
Diarrhea	91 (95.8)	39 (56.5)	<0.001
Abdominal pain	83 (87.4)	39 (55.7)	<0.001
Hematochezia	22 (23.2)	8 (11.4)	0.054
Cough	4 (4.2)	12 (17.1)	0.006
Rhinorrhea	7 (7.4)	15 (21.7)	0.008
Sputum	3 (3.2)	10 (14.3)	0.009
Sore throat	4 (4.2)	0 (0.0)	0.138
Laboratory finding, median (IQR)			
WBC count, cells/μL	9900 (6900–12,850)	9600 (6700–12,950)	0.765
ANC, cells/μL	7050 (4906–10,042)	5588 (3234–10,946)	0.215
ALC, cells/μL	1380 (1045–1730)	1970 (938–3661)	0.019
ESR, mm/h	17 (11–23)	5 (2–13)	<0.001
CRP, mg/dL	6.53 (3.04–9.89)	0.56 (0.21–2.47)	<0.001

IQR: interquartile range; AGE: acute gastroenteritis; URI: upper respiratory infection; LRI: lower respiratory infection; GI: gastrointestinal; FWLS: fever without localizing signs; CNS: central nervous system; WBC: white blood cell; ANC: absolute neutrophil count; ALC: absolute lymphocyte count; ESR: erythrocyte sedimentation rate; CRP: C-reactive protein.

**Table 5 children-08-00538-t005:** Multivariate analysis for significant factors associated with the identification of invasive pathogens.

Factor	Odds Ratio	95% Confidence Interval	*p*-Value
Age, years	1.013	0.892–1.150	0.847
Diagnosis of AGE (vs. non-AGE)	37.846	3.498–409.472	0.003
Fever	13.394	1.350–132.889	0.027
Vomiting	0.417	0.145–1.197	0.104
Diarrhea	5.007	1.050–23.874	0.043
Abdominal pain	2.014	0.492–8.252	0.330
Cough	0.145	0.003–6.807	0.326
Rhinorrhea	1.807	0.113–28.976	0.676
Sputum	9.399	0.382–231.491	0.170
ALC, cells/μL	1.000	1.000–1.000	0.855
ESR, mm/h	1.059	0.997–1.125	0.064
CRP, mg/dL	1.187	1.021–1.379	0.025

AGE: acute gastroenteritis; ALC: absolute lymphocyte count; ESR: erythrocyte sedimentation rate; CRP: C-reactive protein.

## Data Availability

The data presented in this study are available on request from the corresponding author. The data are not publicly available due to national regulations.

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
