# Peer review of "Application of a Multiplex Polymerase Chain Reaction Test for Diagnosing Bacterial Enteritis in Children in a Real-Life Clinical Setting"

_children, 2021, doi:10.3390/children8070538_

Round 1

Reviewer 1 Report

In this manuscript „Application of a Multiplex Polymerase Chain Reaction Test for Diagnosing Bacterial Enteritis in Children in a Real-Life Clinical Setting” of Hyun Woo Lee and colleagues, the authors report on bacterial multiplex PCR (mPCR) for testing patients with clinical symptoms and diagnoses. The mPCR test was performed on >700 patients and the clinical characteristics compared with the mPCR test results. The mPCR test was shown to be significantly associated with some clinical symptoms, which were shown in detail in the manuscript.

The manuscript provides interesting insights into a fast detection system for reliable detection of disease. The manuscript is well written and comprises all necessary information for replication studies. This reviewer has no further comments for an improvement of the manuscript.

Author Response

The authors sincerely thank you for your kind comments.

Reviewer 2 Report

The manuscript by Lee et al. reports the findings of a single-center, retrospective study that examined the clinical features of patients who underwent stool testing with a multiplex PCR panel for bacterial pathogens causing gastrointestinal infections. The study compares patients with positive multiplex PCR panel results to those with negative multiplex PCR results and uses several analyses to identify clinical features associating with the identified pathogen (or lack thereof). The most commonly identified bacterial pathogens detected by PCR included Campylobacter species, Clostridioides difficile, Salmonella species, and Clostridium perfringens. Additional pathogens were detected rarely. Co-detection of two or more pathogens was also fairly frequent. Findings from the study show that patients with positive multiplex PCR identifying a bacterial pathogen were more likely older, had a clinical diagnosis of acute gastroenteritis, had the presence of hematochezia, and absence of cough. The authors further analyzed a subgroup of patients with inflammatory infections (detection of Campylobacter and/or Salmonella) or toxigenic infections (detection of C. difficile and/or C. perfringens). This subgroup analysis demonstrated that patients with inflammatory infections were more likely to have a diagnosis of acute gastroenteritis, have fever and diarrhea, and a higher C-Reactive Protein (CRP) level. The authors concluded that bacterial multiplex PCRs should be best reserved for use in patients presenting with symptoms, and the PCR results interpreted cautiously with consideration of the patients’ clinical symptoms and the clinical disease caused by the identified pathogens.

The study topic is important, although several other publications have utilized various multiplex PCR platforms and reached similar conclusions regarding the need for appropriate diagnostic stewardship in the proper use of these stool studies. The manuscript will likely make a positive, but relatively incremental, contribution to the field. The methods and statistical analysis are appropriate, although I do wish the authors included more information about the age distribution with regards to positive PCR results (see comments below). Tables 2 and 3 in the draft manuscript need some additional work to improve the spacing of the rows and better match the left-hand row titles with the numerical values. The conclusions reached by the authors are supported by the data and results.   

Specific Comments:

  1. Research Design – Should all children less than 19 years of age be included, given the high prevalence of Clostridioides difficile carriage in young children, especially those under 12 months of age? It would be interesting if the authors could report the age distribution of patients with positive testing for Clostridioides difficile to determine how many of these patients were infants and young children.
  2. Results in Tables 2 and 3 – The alignment of several rows are offset, and it is difficult to determine which values correlate with the row labels in the left-hand column. The spacing in the tables must be adjusted to allow for easier reading.
  3. More clarification for the comparison of Inflammatory vs. Toxigenic Pathogens should be included. I found this comparison to be somewhat confusing. Are bacteria producing toxins not inflammatory? Should coli O157/H7 be included in the toxigenic group? Why only include Campylobacter and Salmonella in the inflammatory group and only C. difficile and C. perfringens in the toxigenic group? If this was simply due to insufficient numbers of the other pathogens, please explain this in the methods…lines 114-118 could be more specific in explaining this distinction between the inflammatory and toxigenic group.
  4. Patients identified with inflammatory pathogens (Campylobacter and Salmonella) were found to have a higher likelihood of being diagnosed with acute gastroenteritis, have a fever, have diarrhea, and have a higher CRP than those with toxigenic pathogens ( difficile and C. perfringens). Additional interpretation and discussion of this result could include the fact that asymptomatic carriage of C. difficile and C. perfringens is much more common than the asymptomatic carriage of Campy and Salmonella. Again, the age distribution of patients testing positive for C. difficile, in particular, would be important to highlight as infants and young children have a high rate of asymptomatic carriage which could skew the results.  
  5. English language – There should be at least a moderate review of the use of the English language in this manuscript to improve the reader’s understanding.
    1. There are instances where entire words appear to be missing. For example, line 11…” This study aimed to determine the subjects to bacterial multiplex polymerase chain reaction….”
    2. Another example, Line 48….” This situation may cause a confusion…” Better to say the situation is confusing or may cause confusion due to….

Author Response

The manuscript by Lee et al. reports the findings of a single-center, retrospective study that examined the clinical features of patients who underwent stool testing with a multiplex PCR panel for bacterial pathogens causing gastrointestinal infections. The study compares patients with positive multiplex PCR panel results to those with negative multiplex PCR results and uses several analyses to identify clinical features associating with the identified pathogen (or lack thereof). The most commonly identified bacterial pathogens detected by PCR included Campylobacter species, Clostridioides difficile, Salmonella species, and Clostridium perfringens. Additional pathogens were detected rarely. Co-detection of two or more pathogens was also fairly frequent. Findings from the study show that patients with positive multiplex PCR identifying a bacterial pathogen were more likely older, had a clinical diagnosis of acute gastroenteritis, had the presence of hematochezia, and absence of cough. The authors further analyzed a subgroup of patients with inflammatory infections (detection of Campylobacter and/or Salmonella) or toxigenic infections (detection of C. difficile and/or C. perfringens). This subgroup analysis demonstrated that patients with inflammatory infections were more likely to have a diagnosis of acute gastroenteritis, have fever and diarrhea, and a higher C-Reactive Protein (CRP) level. The authors concluded that bacterial multiplex PCRs should be best reserved for use in patients presenting with symptoms, and the PCR results interpreted cautiously with consideration of the patients’ clinical symptoms and the clinical disease caused by the identified pathogens.

The study topic is important, although several other publications have utilized various multiplex PCR platforms and reached similar conclusions regarding the need for appropriate diagnostic stewardship in the proper use of these stool studies. The manuscript will likely make a positive, but relatively incremental, contribution to the field. The methods and statistical analysis are appropriate, although I do wish the authors included more information about the age distribution with regards to positive PCR results (see comments below). Tables 2 and 3 in the draft manuscript need some additional work to improve the spacing of the rows and better match the left-hand row titles with the numerical values. The conclusions reached by the authors are supported by the data and results.

Specific Comments:

  1. Research Design – Should all children less than 19 years of age be included, given the high prevalence of Clostridioides difficile carriage in young children, especially those under 12 months of age? It would be interesting if the authors could report the age distribution of patients with positive testing for Clostridioides difficile to determine how many of these patients were infants and young children.

Response)

Thank you for your advice.

The age distribution of patients of the toxigenic pathogen group was added in the Table 4 (former Table 3). Patients aged < 2 years were significantly more in the toxigenic pathogen group than in the invasive pathogen group (former invasive group). The median age and proportion of patients aged < 2 years were additionally described in the “Results” section as follows (line 163–165).

For the 44 patients with positive results for C. difficile, the median age was 3 years (IQR: 1–8), and 14 (31.8%) were aged < 2 years.

In the “Discussion” section, the proportion of patients aged < 2 years in those with positive mPCR test results were described, and recommendations for C. difficile testing considering the patient age by the IDSA were described as follows (line 224–234).

The colonization rate of C. difficile is higher in infants and young children than in adolescents and adults, and by 3 years of age, it decreases and becomes similar to that of adults [18]. C. difficile testing should not be routinely recommended for infants with diarrhea and for children aged 1–2 years without the exclusion of other causes of diarrhea [19]. Even for children aged ≥ 2 years, C. difficile testing is recommended for children with risk factors or exposure history to C. difficile infection [19]. In the positive mPCR test group of this study, the positive rate of C. difficile was significantly higher in children aged < 2 years than in those aged ≥ 2 years (51.5% vs. 21.1%, p < 0.001). Moreover, approximately one-third of patients in whom C. difficile was detected were aged < 2 years, and less than half of them had risk factors for C. difficile infection.

  1. Results in Tables 2 and 3 – The alignment of several rows are offset, and it is difficult to determine which values correlate with the row labels in the left-hand column. The spacing in the tables must be adjusted to allow for easier reading.

Response)

Thank you for your kind comments.

To improve readability, the results of multivariate analyses were described in a separate table in the revised manuscript: Table 2 was divided into Table 2 and 3, and Table 3 was divided into Tables 4 and 5. The row and column of each table were re-arranged.

  1. More clarification for the comparison of Inflammatory vs. Toxigenic Pathogens should be included. I found this comparison to be somewhat confusing. Are bacteria producing toxins not inflammatory? Should E. coli O157/H7 be included in the toxigenic group? Why only include Campylobacter and Salmonella in the inflammatory group and only C. difficile and C. perfringens in the toxigenic group? If this was simply due to insufficient numbers of the other pathogens, please explain this in the methods…lines 114-118 could be more specific in explaining this distinction between the inflammatory and toxigenic group.

Response)

Thank you for your valuable comments.

As you indicated, cytotoxin and enterotoxin can cause inflammation in the gut mucosa. Therefore, dividing into “inflammatory” and “toxigenic” can be confusing. Therefore, we re-categorized the bacteria into “invasive” and “toxigenic” pathogens based on the bacterial pathogenic mechanisms described in the Nelson Textbook of Pediatrics. The following sentences were added in the “Results” section with a new citation of the Nelson Textbook of Pediatrics (reference No. 14) (line 118–122).

All the “inflammatory pathogen” was changed to “invasive pathogen” through the whole manuscript.

Excluding rarely identified pathogens, the most frequently identified four pathogens were categorized into invasive and toxigenic pathogens based on their pathogenic mechanisms [14]. Patients with Campylobacter spp. and Salmonella spp. were categorized in the invasive pathogen group (n = 95), and those with C. difficile and C. perfringens were categorized in the toxigenic pathogen group (n = 70).

As you commented, pathogens identified in a small number of patients were excluded from the comparison between the invasive and toxigenic pathogen groups. The following sentence was added in the “Methods” section to describe this intention (line 73–75).

Pathogenic bacteria identified in a small number of patients (< 1% of the whole study population) were not categorized into the invasive pathogen or toxigenic pathogen groups and were excluded from statistical analyses.

  1. Patients identified with inflammatory pathogens (Campylobacter and Salmonella) were found to have a higher likelihood of being diagnosed with acute gastroenteritis, have a fever, have diarrhea, and have a higher CRP than those with toxigenic pathogens (C. difficile and C. perfringens). Additional interpretation and discussion of this result could include the fact that asymptomatic carriage of C. difficile and C. perfringens is much more common than the asymptomatic carriage of Campylobacter and Salmonella. Again, the age distribution of patients testing positive for C. difficile, in particular, would be important to highlight as infants and young children have a high rate of asymptomatic carriage which could skew the results.

Response)

Thank you for your valuable advice.

As responded in the first comment, the age distribution in patients of the invasive and toxigenic pathogen groups were described in the Table 4, and the proportion of patients aged < 2 years were additionally described in the “Results” section as follows (line 163–165).

For the 44 patients with positive results for C. difficile, the median age was 3 years (IQR: 1–8), and 14 (31.8%) were aged < 2 years.

In the “Discussion” section, the proportion of patients aged < 2 years in those with positive mPCR test results were described, and recommendations for C. difficile testing considering the patient age by the IDSA were described as follows (line 224–234).

The colonization rate of C. difficile is higher in infants and young children than in adolescents and adults, and by 3 years of age, it decreases and becomes similar to that of adults [18]. C. difficile testing should not be routinely recommended for infants with diarrhea and for children aged 1–2 years without the exclusion of other causes of diarrhea [19]. Even for children aged ≥ 2 years, C. difficile testing is recommended for children with risk factors or exposure history to C. difficile infection [19]. In the positive mPCR test group of this study, the positive rate of C. difficile was significantly higher in children aged < 2 years than in those aged ≥ 2 years (51.5% vs. 21.1%, p < 0.001). Moreover, approximately one-third of patients in whom C. difficile was detected were aged < 2 years, and less than half of them had risk factors for C. difficile infection.

For C. perfringens, limited reliability of a PCR test for diagnosing C. perfringens infection arisen from its gut colonization was described in the “Discussion” with a new citation (reference No. 20). In addition, comments on the colonization rate of Clostridium spp., Salmonella spp. and Campylobacter spp. were added with new citations (reference No. 21–23) in the “Discussion” section as follows (line 239–250). 

A recent meta-analysis reported low sensitivity (31%) and positive predictive value (49%) of nucleic acid amplification tests against culture for diagnosing C. perfringens-associated diseases [20]. Considering that Clostridium spp. are more likely to exist as intestinal flora than Campylobacter spp. and Salmonella spp. [19,21–23], determining whether the bacteria detected by an mPCR test are true pathogens is more difficult in cases of Clostridium spp. than in cases of invasive pathogens. Furthermore, other bacteria were co-detected in 36.6% of the C. perfringens-positive patients in this study. Therefore, universal inclusion of Clostridium spp. in bacterial mPCR testing seems to be unnecessary for immunocompetent pediatric patients with GI symptoms. Besides, a high number of C. perfringens organisms are needed to cause GI symptoms; therefore, the usefulness of quantitative PCR tests for C. perfringens should be investigated [20].

  1. English language – There should be at least a moderate review of the use of the English language in this manuscript to improve the reader’s understanding.
  2. There are instances where entire words appear to be missing. For example, line 11…” This study aimed to determine the subjects to bacterial multiplex polymerase chain reaction….”
  3. Another example, Line 48….” This situation may cause a confusion…” Better to say the situation is confusing or may cause confusion due to….

Response)

The revised manuscript was reviewed by a professional English-editing company.

The certificate was attached.